# STABILITY REGULARIZATION FOR DISCRETE REPRESENTATION LEARNING

## ABSTRACT

We present a method for training neural network models with discrete stochastic variables. The core of the method is *stability regularization*, which is a regularization procedure based on the idea of noise stability developed in Gaussian isoperimetric theory in the analysis of Gaussian functions. Stability regularization is a method to make the output of continuous functions of Gaussian random variables close to discrete, that is binary or categorical, without the need for significant manual tuning. The method allows control over the extent to which a Gaussian function's output is close to discrete, thus allowing for a continued flow of gradient. The method can be used standalone or in combination with existing continuous relaxation methods. We validate the method in a broad range of settings, showing competitive performance against the state-of-the-art.

## 1 INTRODUCTION

Neural networks are universal approximators of continuous functions. Often, however, discrete computations are desirable, whether for the intermediate neurons and their representations (Oord et al., 2017), the parameters (Courbariaux et al.), or the outputs. Current methods for training neural networks require differentiability which means that it is not straightforward to train neural networks with discrete variables. This has led to the development of several approximate methods (Williams, 1992; Jang et al., 2017; Bengio et al., 2013; Tucker et al., 2017; Pervez et al., 2020) with various trade-offs of bias, variance, and complexity. In this work we focus on neural networks with discrete intermediate representations. Building upon techniques from the analysis of functions in Gaussian spaces (Janson et al., 1997), and specifically the notion of stability of Gaussian functions, we propose a novel regularization strategy on representations that yields precise and hassle-free discrete representations.

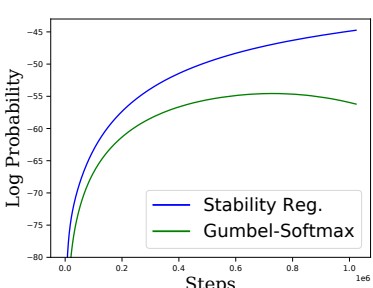

Figure 1: Training curves in MNIST structured prediction with a ResNet model with a 3x7 dimensional binary latent space. Gumbel-Softmax diverges later in training ($\tau = 0.5$).

Several approaches have been introduced in the literature for learning discrete representations with backpropagation. The simplest approach is the Straight-Through estimator (Bengio et al., 2013), which essentially ignores the intermediate discrete function allowing the gradients to flow. Another popular choice is the Gumbel-Softmax (Maddison et al., 2017; Jang et al., 2017), which replaces the discrete categorical variables with relaxed stochastic continuous ones. In both approaches the discrete variables are replaced by approximations and the model is biased with respect to the original discrete objective. When employed with complex architectures, Straight-Through and Gumbel-Softmax estimators often underperform due to this bias, as in figure 1. The reason is that with continuous relaxation methods there is a tension between obtaining better optima and objective function values, and obtaining discrete representations. Importantly, the more complex the optimization (or the model) is the greater as well is the pressure towards non-discrete solutions, thus increasing bias further. Adding to the complexity of obtaining discrete representations, with current methods there is no direct incentive for the optimization procedure to obtain discrete representations: with Gumbel-Softmax the extent of how close-to-discrete a representation is obtained is controlled by a temperature variable, which must be manually tuned.

Unbiased estimators like REINFORCE (Williams, 1992) – and reduced variance extensions like REBAR (Tucker et al., 2017) and RELAX (Grathwohl et al., 2018) – have also been explored. However, these methods tend to be computationally expensive, which limits their usefulness for complex models. All in all, whether due to bias, high variance, high computational complexity, or the need for manual tuning, there remains a need for alternative methods for obtaining hassle-free discrete representations with neural networks, especially when increasing their complexity.

In this work we present a regularization procedure for discrete representations, which can be used either as a standalone method, or in combination with existing continuous relaxation or straight-through estimators. In its standalone form, the method replaces a discrete variable by a parameterized continuous function whose output (say a sigmoid or softmax function) corresponds to the discrete variable, and which is then regularized to produce discrete outputs. In combination with continuous relaxations such as Gumbel-Softmax, the method can be used to regularize the logits input to the sampling procedure to serve as an implicit temperature control by making the logits noise stable.

We achieve this by resting upon the notion of *noise stability* developed in the analysis of Gaussian functions (Borell, 1985; Mossel & Neeman, 2012). Roughly speaking, the noise stability of a Gaussian function is a measure of its resilience to noise. Given a Gaussian function $f$ and correlated Gaussian variables $\epsilon, \epsilon' \in \mathbb{R}^d$, noise stability of $f$ is defined as $\text{Stab} = \mathbb{E}_{\epsilon, \epsilon'}[f(\epsilon)f(\epsilon')]$. Borell's isoperimetric theorem (Borell, 1985), states that for bounded functions of some fixed volume with range $[0, 1]$, noise stability is maximized by functions that are indicator functions of half spaces. Given that half space indicators maximize noise stability in Gaussian space, we suggest that optimizing stability is a very simple and effective method of transforming Gaussian inputs to binary vectors, thus simplifying the process of obtaining discrete representations.

In summary, we demonstrate how the concept of noise stability can be used to regularize stochastic neural networks with Gaussian variables to train hassle-free neural networks with discrete (Bernoulli or categorical) variables.

In the following, we first give a short introduction to noise stability in Gaussian analysis. We then motivate our proposal for using noise stability to regularize Gaussian functions for learning discrete representations. We validate by experiments in the Neural Variational Inference framework to learning graph structured latent spaces, learning discrete (deterministic) autoencoders, clustering with Gaussian Mixture VAEs, gating ResNets, and structured prediction.

## 2 Noise Stability of Gaussian Functions

### 2.1 Stability and Gaussian Isoperimetry

Noise Stability of a Gaussian function $f : \mathbb{R}^n \to \mathbb{R}$ is defined for a noise parameter $\rho \in (0, 1)$ as

$$\text{Stab}_\rho[f] = \mathbb{E}_{\epsilon, \epsilon'}[f(\epsilon)f(\epsilon')], \tag{1}$$

$$\epsilon' = \rho\epsilon + \sqrt{1 - \rho^2}\epsilon'' \tag{2}$$

where $\epsilon, \epsilon'$ are called $\rho$-correlated Gaussian pairs and $\epsilon, \epsilon'' \sim \mathcal{N}(0, 1)$, are samples from the standard normal distribution. Stability is defined here in terms of standard normal Gaussian variables, but it is easily extended to any distribution of independent Gaussian variables by reparameterization, given the mean and standard deviation. For the special case where $f = 1_A$, the indicator function of a set $A$, stability measures the probability that both $\epsilon$ and $\epsilon'$ remain within $A$.

$$\text{Stab}_\rho[f] = \text{P}[\epsilon \in A \wedge \epsilon' \in A], \tag{3}$$

By Borell's Gaussian isoperimetric theorem (Borell, 1985; Mossel & Neeman, 2012), stability is related to the Gaussian isoperimetric inequality. According to Gaussian isoperimetric inequality, geometric objects with minimum boundary (*i.e.*, surface area) in Gaussian space with fixed Gaussian volume (*i.e.*, $\mathbb{E}[f]$ for $f$ the object's indicator function) are half spaces.

**Theorem 1** (Borell Isoperimetric Theorem (Borell, 1985)). *For fixed $\rho \in (0, 1)$ and $f \in L^2(\mathbb{R}^n)$ in Gaussian space with range $[0, 1]$ and fixed volume $\mathbb{E}[f] = \alpha$, $\text{Stab}_\rho[f]$ is maximized by $f = 1_H$ where $1_H$ is an indicator function of a half space with volume $\alpha$.*

As a consequence, given a parameterized bounded Gaussian function, maximizing the stability makes the function $f$ approach the indicator function of some half space as illustrated in figure 2.

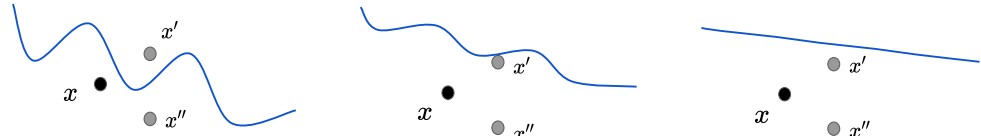

Figure 2: Increasing stability (left to right) makes a function stable to $\rho$-correlated noisy version of $x$ as it approaches a Gaussian halfspace while keeping a fixed Gaussian volume.

For further details on noise stability in the context of Gaussian analysis of functions we refer the interested reader to (O'Donnell, 2014).

## 3 STABILITY REGULARIZATION

Stability regularization can be used either as a standalone method or in combination with Gumbel-Softmax style continuous relaxation.

### 3.1 REGULARIZATION FOR DISCRETE VARIABLES

We start from a continuous model, say a neural network with $L$ layers or modules, $f = f_L \circ \cdots \circ f_1(x)$. Next, we describe how to employ stability regularization so that any arbitrary intermediate function $f_l(z; \theta)$ with bounded output *learns* to output discrete variables. Given input $z$ for $f_l$ we estimate the stability of $f_l$ and, thereafter, maximize it by adding it to the loss objective as a regularizing term.

For a single input vector $z \in \mathbb{R}^k$, following the definition of stability in equation (1), we sample $\rho$-correlated Gaussian variables $\epsilon, \epsilon'$ as in equation (2). We then evaluate $f_l$ twice: once for $z + \epsilon$ and once for $z + \epsilon'$. The expectation of their product, $\mathbb{E}_{\epsilon, \epsilon'}[f_l(z + \epsilon) f_l(z + \epsilon')]$ is the stability, $\text{Stab}_\rho[f_l(z)]$. Maximizing the stability for a single input $z$, $\text{Stab}_\rho[f_l(z)]$ for a fixed $\rho \in (0, 1)$, the function $f_l$ approaches an indicator as described by Borell's theorem.

In a batch setting, we compute a Monte Carlo estimate of the *expected* stability over the input, that is $\mathbb{E}_z[\text{Stab}_\rho[f_l(z)]]$, by sampling *one* $\rho$-correlated Gaussian pair per batch element. Given $z, \epsilon, \epsilon' \in \mathbb{R}^{n \times k}$, so that $\epsilon_i$ and $\epsilon'_i$ are $\rho$-correlated Gaussian, the estimate is computed as

$$\mathbb{E}_z[\text{Stab}_\rho[f_l(z)]] \approx \frac{1}{n} \sum_i f_l(z_i + \epsilon_i) f_l(z_i + \epsilon'_i), \tag{4}$$

where $n$ is the batch size and the arithmetic operations are done element-wise. To maximize stability we sum or average the estimate in equation (4) across dimension and add the result as an additional regularization term to the loss function with which we train the model.

### 3.2 MEAN-CENTERED STABILITY REGULARIZATION

The regularization makes the function stable relative to correlated Gaussian noise by moving the inputs $z_i$ further apart. In some cases the inputs can become too far separated which can hurt optimization if left uncontrolled. For such problematic cases we introduce *mean-centered stability regularization* which preserves the expected value of $f_l$ given the input $z_i$ ensuring that the induced separation remains limited.

The idea behind *mean-centered stability regularization* is compute the stability of $f_l - \mathbb{E}[f_l]$ so that maximizing stability reorients the separating hyperplane without changing the Gaussian volume of the corresponding halfspace. According to Borell's theorem maximizing stability of $f_l$ for *any* fixed $\mathbb{E}[f_l]$ causes it to approach a Gaussian halfspace indicator (figure 2). Since it can be expensive to compute the expectation of a neural network, we maximize the difference of two stability computations: Given parameters $\rho_1, \rho_2, \rho_2 < \rho_1$ we optimize $\text{Stab}_{\rho_1}[f_l] - \text{Stab}_{\rho_2}[f_l]$. We can show that this objective is equal to $\text{Stab}_{\rho_1 - \rho_2}[f_l - \mathbb{E}[f]]$ to first order with an error that is quadratic in $\rho_1, \rho_2$.

**Proposition 1.** *Given a Gaussian function* $f : \mathbb{R}^n \to [0, 1]$ *and parameters* $\rho_1, \rho_2 \in (0, 1)$, $Stab_{\rho_1 - \rho_2}[f - \mathbb{E}[f]] = Stab_{\rho_1}[f] - Stab_{\rho_2}[f] + O(\rho_2(\rho_2 - \rho_1))$.

---

**Algorithm 1** Stability Regularization

---

**Require:** Input $z \in \mathbb{R}^{n \times k}$; stability layer $f_l$ with range $(0,1)^m$; noise parameter $\rho \in (0,1)$; stability constraint $\alpha \in (0,1)$
1: Sample $\epsilon, \epsilon' \in \mathbb{R}^{n \times k}$ $\rho$-correlated Gaussian vectors.
2: Compute $y_1 = f_l(z + \epsilon)$, $y_2 = f_l(z + \epsilon')$
3: Estimate average stability over batch per dimension as $S = \frac{1}{n} \sum_i y_{1,i} y_{2,i}$.
4: Apply stability constraint per dimension: $S = \mathtt{clip}(S, 0, \alpha)$.
5: Sum $S$ across dimensions and optimize by gradient descent

---

**Algorithm 2** Stability Regularization with Mean Centering

---

**Require:** Input $z \in \mathbb{R}^{n \times k}$; stability layer $f_l$ with range $(0,1)^m$; noise parameter $\rho_1, \rho_2 \in (0,1)$, $\rho_2 < \rho_1$
1: Sample $(\epsilon_1, \epsilon_1')$ $\rho_1$-correlated and $(\epsilon_2, \epsilon_2')$ $\rho_2$-correlated Gaussian vectors from $\mathbb{R}^{n \times k}$.
2: Compute $y_1 = f_l(z + \epsilon_1) f_l(z + \epsilon_1')$, $y_2 = f_l(z + \epsilon_2) f_l(z + \epsilon_2')$
3: Estimate average stability over batch per dimension as $S = \frac{1}{n} \sum_i (y_{1,i} - y_{2,i})$.
4: Sum $S$ across dimensions and optimize by gradient descent

---

See appendix A for a proof.

For stability regularization without mean centering we clip the stability at a maximum value to prevent the network output from becoming overly saturated. In figure 2 this would correspond to the points becoming far from the boundary leading to saturation and slowdown of optimization. Without mean centering, a constraint on stability limits how far the points can be from the boundary improving optimization.

Borell's theorem guarantees that the function $f$ will converge to a halfspace for any $\rho$. We did not observe the method to be sensitive to $\rho$ in our experiments.

The precise procedures are described in algorithms 2 and 1.

**Stability Regularized Layers.** The stability regularized neural network layers can be any arbitrary bounded output neural network layer, possibly even a layer of activations without learned parameters. We use sigmoid activations for Bernoulli and softmax for categorical variables.

**Probabilistic Models and Gumbel-Softmax.** We use stability regularization alongside Gumbel noise in probabilistic models such as VAEs where it is important to be able to compute log probabilities of obtained samples. Given a block of layers $f_l$ with a Gumbel softmax (or Gumbel sigmoid) activation, i.e., $f_l = \mathrm{GumbelSoftmax}(\mathrm{logits})$, we compute stability using a standard softmax or sigmoid without adding the Gumbel noise as $\mathrm{Stab}_\rho[\mathrm{Softmax}(\mathrm{logits})]$ and use the Gumbel softmax output as input to the downstream network.

The optimization procedure with continuous relaxations provides no incentive to encourage discrete representation. The consequence is that such methods work better when there is little pressure from the optimization pressure to be non-discrete, as happens with larger latent space dimension. With smaller bottleneck latent spaces, however, there is greater optimization pressure to be continuous and the optimization with continuous relaxations becomes harder because of the need to manually tune the temperature. With stability regularization the regularization procedure is a form of implicit temperature control and the extent of how discrete a representation becomes is controlled by the extent of regularization.

**Computational Complexity.** Stability regularization is easy to implement and adds some extra computations due to the extra evaluations for correlated Gaussians. We emphasize that any extra computation is local to the stability layer $f_l$. The rest of the network is unaffected. Depending on the application, there usually exist only a few such layers in a large model, in which case the stability computation is a small fraction of the total cost and we do not observe a noticeable increase in computational cost in our experiments.

## 4    RELATED WORK

A number of methods have appeared in the literature for training neural networks with discrete variables. The methods can broadly be divided into two categories: score function based methods, and pathwise gradient methods (Mohamed et al., 2019).

Score function based methods are usually unbiased – the prototypical example being REINFORCE (Williams, 1992). The simplest form of these methods has high variance and a number of approaches to reduce the variance of the REINFORCE estimator have appeared. NVIL (Mnih & Gregor, 2014) subtracts a learned MLP baseline from the REINFORCE estimator to reduce variance, while MuProp (Gu et al., 2016) subtracts a sample-dependent baseline based on the first order Taylor series expansion. REBAR (Tucker et al., 2017) combines the REINFORCE estimator with continuous relaxation baseline, while RELAX (Grathwohl et al., 2018) generalizes the REBAR control variate to one parameterized by a neural network. Yin et al. propose ARSM, a finite difference estimator with adaptive evaluation.

Pathwise gradient methods make use of the functional form of the operations that are applied to the discrete random samples, and consequentially tend to have lower variance than score-function methods (Mohamed et al., 2019). However, since gradients through discrete nodes are not defined, approximations have to be made, which makes such estimates biased in general. Continuous relaxation estimators replace discrete nodes with continuous stochastic variables. Gumbel-Softmax (Maddison et al., 2017; Jang et al., 2017) is an example of this, which relaxes the Gumbel-Max parameterization of discrete variables. (Potapczynski et al., 2020) extend Gumbel-Softmax to Gaussian variables.

Straight-Through estimators (Bengio et al., 2013) are another category of biased estimators, which provide hard samples in the forward pass but skip the discrete node in the backward pass. Straight-through estimators can also be combined with continuous relaxation estimators (Jang et al., 2017) to provide hard samples. In DARN (Gregor et al., 2014) develop another pathwise gradient estimator to be unbiased for quadratic functions but is shown to be a lower bias Straight-Through estimator in (Pervez et al., 2020). FouST (Pervez et al., 2020) also employs similar correlated samples but in the context of variance reduction in Bernoulli-input networks.

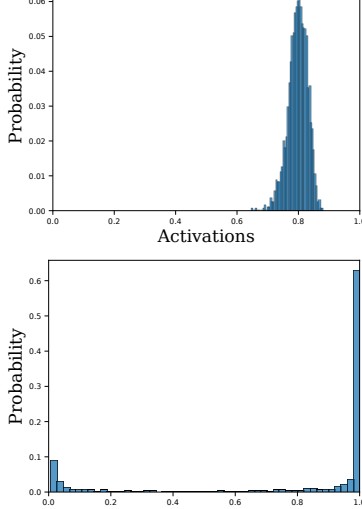

## 5    EXPERIMENTS

We validate our stability regularization procedure on a number of tasks with Bernoulli and categorical variables to show its effectiveness and wide applicability. First, with a simple example we show that stability regularization can indeed be used to learn Bernoulli random variables with Gaussian functions and that we can control the extent of saturation (or of how close a function is to being Boolean) by constraining stability to some fixed upper bound. We then demonstrate experiments on 1) latent structure recovery with Neural Relational Inference (Kipf et al., 2018) 2) autoencoders with Bernoulli variables and generative models for images, 3) Gaussian mixture VAEs for unsupervised clustering and 4) conditional computing, where individual layers of a ResNet model can be dynamically switched off to save computation.

Figure 3: A toy regression example. The Gaussian function's mean output approaches the target probability of 0.8 with and without stability regularization. The histograms show the outputs clustering around the target without regularization (top) and approaching binary values with regularization (bottom) which average to 0.8.

### 5.1    A TOY DISCRETE REGRESSION EXAMPLE

As a simple illustration of stability regularization we train a small two-layer neural network with sigmoid output and standard normal input to output binary values. We train the neural network using a regression objective so that the expected output of the network matches a target $t$, i.e., minimizing $(\mathbb{E}[f(z)] - t)^2$ with and without stability regularization on $f$. The results for a run with a target of

Table 1: Categorical VAE experiment with stability regularization showing negative best validation ELBO (best of 5 runs) where $\tau$ is the Gumbel-softmax temperature.

| Method | Gumbel-Softmax | | | | | Straight-Through |
|---|---|---|---|---|---|---|
| | $\tau = 0.1$ | $\tau = 0.5$ | $\tau = 1$ | $\tau = 1.5$ | $\tau = 2$ | |
| Without Stab Reg. | 107 | 107.7 | 114 | 122.7 | 132.4 | 116.7 |
| With Stab Reg. | 107.4 | 106.8 | 112.8 | 116.8 | 122.7 | 111.9 |

0.8 is shown in figure 3. As shown, without regularization the outputs cluster around the target, as is expected with regression, while with regularization the network outputs binary values with the mean output matching the regression target.

## 5.2 CATEGORICAL VAE

To judge the effect of stability regularization we perform experiments with categorical VAE on dynamically binarized MNIST. We train the VAE using Gumbel-Softmax and Straight-Through both with and without stability regularization. The aim is to show that traditional gradient estimators benefit from stability regularization and that stability regularization can be interpreted as performing implicit temperature control when used with continuous relaxation.

We use VAE with the following architecture. The encoder has one layer of 200 units and the ELU non-linearity. The decoder has one layer of 400 units followed by the ELU non-linearity. The stochastic layer consists of 100 categorical variables with 2 categories and is implemented as a single layer of 200 units computing the logits for each of the categories. For stability regularization we use exactly the same architecture; in particular we do not use a separate stability layer and regularization is performed directly on the logits. This is a restriction but has the advantage that the methods remain directly comparable and also that the KL divergences are easily computable because of independence.

Table 2: Regularization coefficient with categorical VAE with Gumbel-Softmax at $\tau = 2$

| Value | -V. ELBO |
|---|---|
| 80 | 124.6 |
| 100 | 122.7 |
| 120 | 122.2 |
| 150 | 123.1 |
| No Reg. | 132.4 |

We train all experiments for 500 epochs. The input is binarized in each batch by sampling Bernoulli values independently for each pixel by treating the pixel value as the probability. We train using Adam with a learning rate of 1e-4. For Gumbel-Softmax we use temperatures in $\{0.1, 0.5, 1.0\}$. The stability regularization coefficients are chosen from $\{20, 50, 100\}$. The parameter $\rho$ is set to 0.9. Straight-Through is implemented by sampling 1-hot vectors according to logits and passing the gradient to the softmax (instead of the logits) in the backward pass.

Comparisons with Gumbel-Softmax and Straight-Through with and without stability regularization are shown in Table 1. We see that stability regularization improves the Straight-Through trained model significantly ($\sim$5 nats) and the Gumbel-Softmax trained models when the temperature is higher ($\sim 10$ nats at $\tau = 2$). This shows evidence for our suggestion that stability regularization provides implicit temperature control with continuous relaxation.

To study the effect of the regularization term coefficient on training we show validation ELBO for categorical VAE trained with Gumbel-softmax with fixed temperature set to 2 and stability regularization with various coefficient values in Table 2. Coefficient values are chosen from $\{80, 100, 120, 150\}$. The table shows that training performance is relatively stable (within $\pm 1$ nats) across a range of coefficient values. Compared with the range of ELBO values obtained when changing the temperature in Table 1, we conclude that the action of the regularization coefficient is different from that of the temperature.

## 5.3 GRAPH STRUCTURED LATENT SPACES

We perform experiments with Neural Relational Inference (Kipf et al., 2018), which is a variational autoencoder model to recover interactions and learn dynamics given observational data. The latent space of the NRI VAE model is graph structured, and the encoder and decoder are graph neural networks. To obtain sample interaction graphs in the latent space, NRI models use Gumbel-Softmax to sample edges producing a graph sample and to propagate gradients through the latent graph. For the NRI experiments, we use stability regularization along with Gumbel-Softmax.

**Physics Simulation Experiments** The first set of NRI experiments are physics simulations from the original NRI proposal Kipf et al. (2018). This has three types of systems of 1) particles connected by springs, 2) charged particles and 3) phase-coupled oscillators with the Kuramoto model (Kuramoto, 1975). Synthetic data was generated using the authors' code with 50k training and 10k validation and test samples each. For each type of system we have further two types of experiments with either 5 or 10 objects.

Table 3: NRI Physics Simulation Accuracy. * indicates result different from number reported.

| Model | Springs | Charged | Kuramoto |
|---|---|---|---|
| | | 5 Objects | |
| Supervised | $99.9_{\pm 0.0}$ | $95.0_{\pm 0.3}$ | $99.7_{\pm 0.0}$ |
| NRI-GS | $99.9_{\pm 0.0}$ | $82.1_{\pm 0.6}$ | $93.9*$ |
| NRI-GS+Stab | $99.9_{\pm 0.0}$ | $88.1_{\pm 0.2}$ | $95.3_{\pm 0.1}$ |
| | | 10 Objects | |
| Supervised | $98.8_{\pm 0.0}$ | $94.6_{\pm 0.2}$ | $97.1_{\pm 0.1}$ |
| NRI-GS | $98.4_{\pm 0.0}$ | $70.8_{\pm 0.4}$ | $66.5*$ |
| NRI-GS+Stab | $98.4_{\pm 0.0}$ | $75.0_{\pm 1.0}$ | $68.7_{\pm 0.6}$ |

We compare the NRI physics experiments with and without stability regularization. For this experiment we used the same baseline model with no new parameters and the stability regularization is used with a softmax applied to the logit parameters going into the Gumbel-Softmax. For this experiment we used stability regularization without mean centering, a $\rho$ value of 0.9, and a constraint of 0.9 for the sum of stability across the softmax dimensions. We add the stability loss to the optimization objective with a multiplied by a factor of 100.

For the 5 and 10 particles springs experiments, we match the NRI baseline, which already matches the supervised baseline. For the 5 and 10 charged particles experiments we get a significant improvement with stability regularization, achieving an accuracy improvement of about 6% and 5%.

For the Kuramoto oscillator experiments, we achieve accuracy improvements of 1.5% to 2% over 5 and 10 oscillator experiments. Here we note that the originally reported accuracy values for the Kuramoto model are higher (96% and 75.7%) but we were unable to reproduce the numbers in our experiments. The Kuramoto values we report were obtained by running the provided code against which we compare our method. We note that the improvements are obtained solely by including stability regularization without model changes.

Table 4: Comparing with Paulus et al. (2021) in latent spanning tree recovery with 3,5 and 10 steps, w.r.t. test ELBO, precision, recall.

| Method | ELBO | Pr | Re |
|---|---|---|---|
| | T=10 | | |
| Top$|V|$-1 | -2100 | 41 | 41 |
| Top$|V|$-1+Stab | -1766 | 92 | 92 |
| SpTree | -1080 | 91 | 91 |
| SpTree+Stab | -1175 | 89.3 | 89.3 |
| | T=5 | | |
| SpTree | -516 | 82.3 | 82.3 |
| SpTree+Stab | -501 | 82.6 | 82.6 |
| | T=3 | | |
| SpTree | -221 | 65.9 | 65.9 |
| SpTree+Stab | -196 | 70.52 | 70.52 |

**Latent Tree Structure Recovery** We perform further experiments with NRI on latent spanning tree recovery proposed by Paulus et al. (2021). Here a dataset is generated by sampling a tree over 10 vertices, embedding the vertices and applying a force directed graph drawing algorithm (Fruchterman & Reingold, 1991). The dataset consists of particle locations at each step obtained by applying the drawing algorithm for $T$ steps. Paulus et al. (2021) develop a NRI method for sampling tree structured latent spaces and show improved performance with structure recovery compared with sampling independent edges as in the baseline NRI model and Top $|V| - 1$ selection where V is the vertex set. Their conclusion is that latent tree sampling works better than the baseline for T=10 iterations.

Using the authors' code we performed experiments with shorter T=3,5 and T=10 step trajectories with the spanning tree and Top $|V| - 1$ sampling with and without stability regularization. In general, recovering the interaction structure is harder when shorter trajectories are observed. We used mean centered stability regularization with a one layer stability network, which we took as the last linear layer of the encoder network. We used $\rho_1 = 0.9, \rho_2 \in \{0.89, 0.8, 0.5\}$. We ran random search over the Gumbel-Softmax temperature and learning rate and ran the experiments for multiple random seeds for the best hyperparameters found by the search.

We first focus on the spanning tree methods, which resemble the true latent structure of the data. Including stability regularization matches performance at T=5,10 steps, and improve with T=3 steps

from around 66% to 70%. What is more interesting, when focusing on the Top $|V| - 1$ method for $T = 10$, we find that having stability regularization we can match the spanning tree method with over 90% precision and recall. This is remarkable considering that the latent space is that of a spanning tree, and that using a Top $|V| - 1$ without regularization from Paulus et al. (2021) scores a 41% accuracy. When reducing the number of steps, however, the Top $|V| - 1$ does not have the capacity to infer the latent structure, attaining only a low precision and recall of 35% for the best validation score.

For this experiment we find that stability regularization made the method significantly less sensitive to hyperparameters and random seed. We conclude that with stability regularization helps in the low data limit for this task or when the latent space has less structure than the ground truth alongside making the method less sensitive to hyperparameters and random seed.

## 5.4 Unsupervised Clustering with Gaussian Mixture VAEs

To show an application of stability regularization to categorical variables we turn to Gaussian mixture VAEs and use them for unsupervised clustering for MNIST and OMNIGLOT. We use a very simple generative model where the mean and variance of the Gaussian mixture components are functions of a sampled categorical variable. In particular, we do not marginalize over the categorical variable and resort to sampling only.

$$b \sim p(b) = \text{Categorical}(\eta)$$
$$z \sim p(z|b) = \mathcal{N}(\mu_{\theta_1}(b), \sigma_{\theta_2}(b))$$
$$x \sim p(x|z)$$

where $\mu_{\theta_1}, \sigma_{\theta_2}$ are MLP networks with categorical input. We use the following approximate posterior.

$$z \sim q(z|x) = \mathcal{N}(\mu_{\phi_1}(x), \sigma_{\phi_2}(x))$$
$$b \sim q(b|z) = \text{Categorical}(g_{\phi_3}(z))$$

Table 5: Unsupervised classification accuracy for MNIST

| Model | Accuracy |
|---|---|
| AAE (Makhzani et al.) | 95.9 |
| GMVAE (Dilokthanakul et al.) | 92.7 |
| SB-VAE (Nalisnick & Smyth) | 92.35 |
| Stab. Reg. (20 Clusters) | 93.3 |

The networks in $p(x|z), q(z|x)$ are ResNets with two ResNet blocks, while $g_{\phi_3}$ is implemented as an MLP. The categorical variable is implemented as a softmax at the output of a stability network in $g_{\phi_3}$, which we regularize to produce categorical outputs. Computing KL divergence requires probabilities for the categories, which we approximate using a separate two-layer MLP with softmax output, trained to regress over the categorical outputs. To train the model we maximize the ELBO plus the auxiliary stability and regression objectives. We *do not* use Gumbel Softmax and use standalone stability regularization for the categorical variables.

We obtain an average unsupervised classification accuracy on MNIST of 93.3% (see table 5). Label assignment is done using the label of the maximum probability example in a cluster. We observe that while relying on a simple model, we obtain classification accuracy that is better than or close to more complex models from the state-of-the-art. We provide generation examples from 15 clusters for MNIST and OMNIGLOT in figure 7 in the appendix.

## 5.5 Gating ResNets

As an application of Boolean variables to gating, we consider the task of selectively turning off layers of a deep ResNet using gates. We chose the *adaptive inference network* architecture, which builds on a ResNet-110 architecture for CIFAR-10 (Veit & Belongie, 2017). The architecture incorporates Boolean variables to serve as gates, which are trained using Gumbel-Softmax. The architecture also allows setting a target rate at which individual layers can be turned off.

We use the authors' PyTorch implementation[1] and replace the Gumbel-Softmax training with *standalone* stability regularization. We use the average activation over the batch as a probability estimate in algorithm 1 and stop stability gradients at the stability block input. We use the default settings of parameters found in the implementation. With target rates of 0.5 and 0.6 we were able to obtain

---

[1]https://github.com/andreasveit/convnet-aig

accuracy values of 92.86 and 93.31 for the baseline implementation with Gumbel-Softmax training. For our stability regularization method we obtain accuracy values of 93.2 and 93.46 with the rate set to 0.5 and 0.6. While the improvements are modest, the already high accuracies makes any further increase hard. With additional tweaks, Veit & Belongie (2017) report a higher accuracy of 94.24. In the above experiments, however, we use the published implementation with the default hyperparameters. We conclude that stability regularization can also be deployed for dynamically switchable layers in deep architectures.

### 5.6 DISCRETE AUTOENCODERS AND GENERATIVE MODELING FOR IMAGES

As an application of stability regularization, we train discrete autoencoders on natural images solely with binary latent variables, where the encoder and decoder are deep ResNets. We use a Gaussian output model for the decoder. After training the discrete autoencoder we train a PixelCNN prior over the binary latent variables to get a generative model. This is similar to the setup used by Oord et al. (2017), with the difference that we do not quantize, instead we use *standalone* stability regularization to obtain a discrete latent space.

To generate new samples, we first sample a binary latent variable vector from the PixelCNN prior model, which is then fed to the decoder to generate samples. We train the model on CIFAR-10 with 50k 32x32 labeled images and STL-10 with 100k 96x96 unlabeled images. For CIFAR-10 and STL-10 we use binary latent spaces of dimensions 16x16x4 and 24x24x4 respectively. This corresponds to a bit-size reduction of $\frac{32 \times 32 \times 3 \times 8}{16 \times 16 \times 4} = 24$ times for CIFAR-10 and 96 times for STL-10.

Sampled images from the models trained on CIFAR-10 and STL-10 are shown in figure 4 and B.1 in the appendix. We also train a conditional PixelCNN prior model for CIFAR-10 by using the CIFAR-10 training labels. Samples from the conditional model are shown in figure 4. The generations show that the model is able to capture global information better than a PixelCNN model trained on raw images. For the CIFAR-10 unconditional model we obtain an FID score of 64.3 which is a slight improvement over the value reported by Ostrovski et al. for a PixelCNN model trained on raw images. For the conditional CIFAR-10 model we obtain an FID score of 54.3. A comparison of FID scores for various models on CIFAR-10 is in table 6 in the appendix. The FID scores were obtained on a highly compressed representation with a standard PixelCNN. The scores can likely be improved by using lower compression or an improved PixelCNN but we do not explore this in this work.

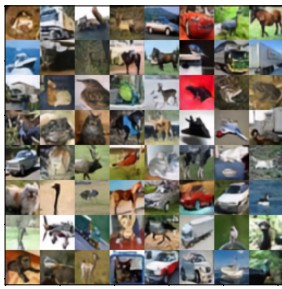
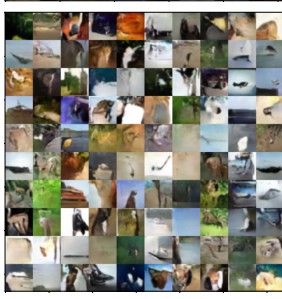
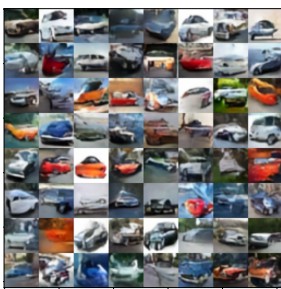

Figure 4: CIFAR-10 reconstructions with a 16x16x4 binary latent space (top); generated samples with a pixelCNN prior on the latent space per class (middle); and conditional samples from a single class (bottom)

## 6 CONCLUSION

We presented a new method for regularizing stochastic Gaussian neural networks, so that to train accurately and hassle-free models with Boolean and categorical stochastic variables. For this, we rely upon the notion of *noise stability* developed in the analysis of Gaussian functions, which is maximized by functions that are indicator functions of half spaces. We validate successfully stability regularization on a wide array of experiments and settings, where Boolean and categorical random variables are required, including physics simulations with graph latent variable models, Gaussian mixture models for clustering, gating applications for large neural networks, and autoencoders. Importantly, we find that stability regularization requires limited tuning compare to other continuous relaxation methods, making it a strong contender for models with discrete variables in practice.

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

# A   PROOF OF PROPOSITION 1

*Proof.* We use the following expression of stability in terms of the Hermite expansion of $f$.

$$\text{Stab}_\rho[f] = \sum_\alpha \rho^{|\alpha|} \hat{f}(\alpha)^2, \tag{5}$$

where $\alpha$ is a multi-index indexing the Hermite polynomial basis $h_\alpha$, and $\hat{f}(\alpha)$ is the coefficient of $f$ in the Hermite expansion of $f$ corresponding to the basis function $h_\alpha$ (O'Donnell, 2014).

Given parameters $\rho_1, \rho_2\ \rho_2 < \rho_1$, we have

$$\text{Stab}_{\rho_1 - \rho_2}[f - \mathbb{E}[f]] = \sum_{\alpha:|\alpha|>0} (\rho_1 - \rho_2)^{|\alpha|} \hat{f}(\alpha)^2, \tag{6}$$

where we use the fact that $\mathbb{E}[f] = \hat{f}(0)$. Also

$$\text{Stab}_{\rho_1}[f] - \text{Stab}_{\rho_2}[f] = \sum_{\alpha:|\alpha|>0} (\rho_1^{|\alpha|} - \rho_2^{|\alpha|}) \hat{f}(\alpha)^2 \tag{7}$$

$$= \text{Stab}_{\rho_1 - \rho_2}[f - \mathbb{E}[f]] + O(\rho_2(\rho_2 - \rho_1)) \tag{8}$$

$\square$

# B   FURTHER EXPERIMENTAL DETAILS

## B.1   DISCRETE AUTOENCODERS

For CIFAR-10 we downsample once using strided convolutions in the encoder. After the downsampling the encoder has 3 ResNet blocks with 80 feature maps. This is followed by the stability network which has two convolutions layers followed by two ResNet blocks all with 80 features maps followed by a convolutional layer with 4 features maps. We upsample using transposed convolutions in the decoder followed by 3 ResNets blocks.

For STL-10 we downsample twice using strided convolutions in the encoder. After the first downsampling we use one convolutional layer followed by a ResNet block. After the second downsampling we use 6 ResNet block. This is followed by the stability network which has two convolutions layers followed by two ResNet blocks all with 80 features maps followed by a convolutional layer with 4 features maps. We use 3 ResNet blocks with 80 feature maps. The first upsampling operation in the decoder followed by 7 ResNets blocks with 80 features maps. The second upsampling operation in the decoder followed by 2 ResNets blocks with 64 features maps.

Table 6: FID scores for CIFAR-10

| Model | FID |
|---|---|
| PixelCNN (Ostrovski et al.; Oord et al., 2016) | 65.93 |
| PixelIQN (Ostrovski et al.) | 49.46 |
| DCGAN (Radford et al.; Arjovsky et al., 2017) | 37.11 |
| NVAE (Vahdat & Kautz, 2020; Aneja et al.) | 51.10 |
| Discrete AE+Stab. Reg. | 64.39 |
| Discrete AE+Stab. Reg. (cond.) | 54.40 |

The output model in both cases is Gaussian.

We use $\rho \in \{0.8, 0.9, 0.95\}$ and a stability constraint of $0.6$. To evaluate we threshold the output of the stability network at $0.5$ so that greater values become 1 and the rest 0. We train with Adam with a learning rate of 8e-5.

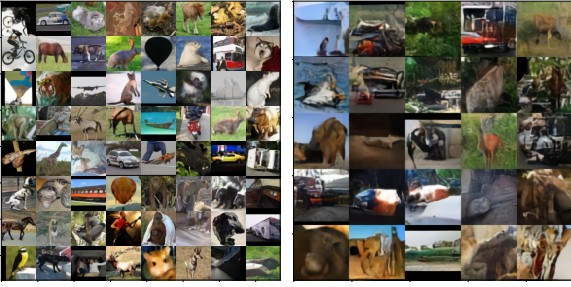

Figure 5: STL-10 reconstructions with a 24x24x4 binary latent space (left); generated samples with a pixelCNN prior on the latent space (right).

## B.2 STRUCTURED PREDICTION

As further validation of stability regularization to train models with Boolean variables we experiment with the MNIST structured prediction task where the goal is to generate one half of the image given the other half. This task was used in Raiko et al. (2014); Tucker et al. (2017) to test and compare various gradient estimators for Boolean latent variables. We use the *dynamically binarized* dataset for this task. We use an MLP architecture we 2 layers of 200 tanh units in the encoder and decoder and similarly a stability block with 2 layers of 200 tanh units. The output of the encoder is fed to the stability block and the output of the stability block goes into the encoder. To evaluate we threshold the output of the stability block so that the decoder only sees Boolean values during evaluation. We compare validation log likelihood curves with a single sample against Gumbel-Softmax, Rebar and MuProp for which we use code from the Rebar code repository. Since our model has more parameters due to the stability block, we use encoder and decoder architectures of 250 tanh units for these models to make the comparison fairer. We show validation curves in figure 6.

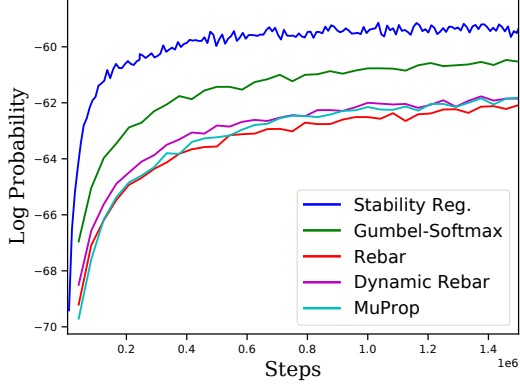

Figure 6: Validation curves for structured prediction on dynamic MNIST

We find that, in terms of validation likelihood, Rebar performs worse than Gumbel-Softmax with the dynamic MNIST dataset on this task while stability regularization outperforms all others by a large margin.

## B.3 FURTHER ABLATIONS

### B.3.1 NOISE PARAMETER

We study the effect of the noise parameter $\rho$ on VAE performance. Using the same setup as in Section 5.2 and using Gumbel-Softmax with a fixed temperature of 1, we train models with $\rho$ parameters in $\{0.5, 0.6, 0.8, 0.9\}$. We run the experiment for 300 epochs instead of 500 as in Section 5.2. The results shown in Table 7 indicate that the performance is minimally affected by varying choices of $\rho$ all runs achieving a validation ELBO of about -113.8.

Table 7: Negative validation ELBO when varying $\rho$ with categorical VAE with Gumbel-Softmax at $\tau = 1$

| $\rho$ | -V. ELBO |
|--------|----------|
| 0.5 | 113.8 |
| 0.6 | 113.9 |
| 0.8 | 113.8 |
| 0.9 | 113.7 |

### B.3.2 GUMBEL NOISE IN REGULARIZATION

When computing the stability regularization objective alongside Gumbel-Softmax we evaluate the stability layer without Gumbel noise in the experiments, using a plain softmax as the output of the stability layer. Gumbel noise is still used for the input to the downstream decoder network.

Here we study the effect of using Gumbel Noise also in the stability computation. We use the same setup as in Section 5.2 and use stability regularization with Gumbel noise for various temperatures. The results are shown in Table 8. Here we find that for stability computation with Gumbel noise we get an improvement of about 2 nats for higher temperatures $\tau = 1.5, 2$. Whereas for lower temperatures we get on-par or marginally worse performance. We conclude that under specific conditions Gumbel noise in regularization can lead to a small benefit, but the overall effect is marginal.

Table 8: Categorical VAE experiment with stability regularization using Gumbel noise for stability. Showing negative validation ELBO.

| Method | Gumbel-Softmax | | | | |
|--------|----------------|----------------|--------------|----------------|--------------|
| | $\tau = 0.1$ | $\tau = 0.5$ | $\tau = 1$ | $\tau = 1.5$ | $\tau = 2$ |
| Without Stab Reg. | 107 | 107.7 | 114 | 122.7 | 132.4 |
| Stab Reg. | 107.4 | 106.8 | 112.8 | 116.8 | 122.7 |
| Stab Reg.+Gumbel Noise | 107.4 | 107.3 | 112.9 | 114.4 | 120.1 |

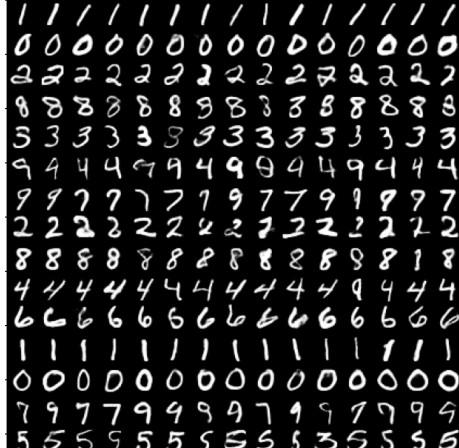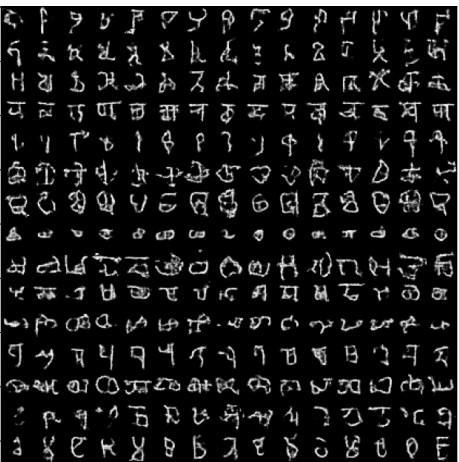

Figure 7: Unsupervised clustering with a Gaussian Mixture VAE trained with stability regularization. The images show 15 clusters (one per row) and corresponding generated images for MNIST (left) and OMNIGLOT (right).

