# OpenReview forum: "Stability Regularization for Discrete Representation Learning"
_ICLR.cc/2022/Conference — ICLR 2022 Poster_

### Official Review · Reviewer_ZtpK · 2021-10-29

**Correctness:** 4
**Technical Novelty And Significance:** 3
**Empirical Novelty And Significance:** 3
**Recommendation:** 6
**Confidence:** 3

**Main Review:**

The paper builds on the concept of noise stability of functions of Gaussian variables.  Noise stability measures the resilience of such functions
to noise by estimating how their outputs correlate when they receive as input correlated Gaussian variables; the higher the correlation the more
stable the function. For a family of functions, bounded and fix volume, the stability is maximized for these functions that are indicator functions
of half spaces which is a natural way to construct binary variables. The paper proposes to maximize stability in order to obtain categorical
variables.

The implementation of the method is rather simple and it comes with a small computational overhead due to the fact that it requires a double evaluation
of these computational units whose output we would want to push to be discrete.

The paper explores the performance of the method in a diverse set of experiments learning discrete latent spaces, gating computational units using binary
variables, and generating discrete outputs. I appreciate the breadth of scenarios, I only have a small comment here on the fact that the comparisons are
only done with the Gumbel-softmax, why this choice and not compare also with some of the other approaches that the authors discuss, e.g. straigh-through
and/or some variants of the score based estimators?



**Summary Of The Paper:**

The paper presents a new method to train neural networks with stochastic discrete variables called stability regularisation. The method pushes
the outputs of functions of Gaussian random variables to be close to discrete, and unlike other methods used with discrete variables,
such as Gumbel Softmax which requires temperature annealing, it is easy to tune. The method is demonstrated on a very rich variety of
tasks and models and shows state of the art performance.


**Summary Of The Review:**

Overall this is a nice contribution to what is an important problem, learning discrete (stochastic) variables. The idea is simple and the authors demonstrate it in a variety of datasets. The only thing that I would have liked to see is a comparison with other methods that are meant to also work in such discrete settings, e.g. straight-through, score-based estimators.

---

> ### Author Response · Authors · 2021-11-17
> **Response**
>
> We thank the reviewer for the appreciative review.
>
> >> I only have a small comment here on the fact that the comparisons are only done with the Gumbel-softmax, why this choice and not compare also with some of the other approaches that the authors discuss, e.g. straigh-through and/or some variants of the score based estimators?
>
> We have added some new experiments on training categorical VAEs in the updated paper in section 5.2. In particular we show a comparison against various estimators including Straight-Through and Gumbel-Softmax at various temperatures. The results show that using straight-through with stability results in an improvement of about 5 nats. Also the method helps Gumbel-Softmax more in the high temperature regime.
>
> In the paper we mainly concentrate on pathwise gradient estimators since these are more efficient and are easier to scale to larger models. However, we do compare against score function methods (Rebar and MuProp) on MNIST structured prediction with dynamically binarized MNIST (please see appendix B.2), and find that the score function methods have worse scores on the validation set. This is also reported in the Rebar paper [1], where Gumbel-Softmax outperforms other methods in terms of validation score for MNIST structured prediction. We conclude that stability regularization leads to improved performance on this method.
>
> [1] Tucker et al. REBAR: Low-variance, unbiased gradient estimates for discrete latent variable models

---

### Official Review · Reviewer_jxs7 · 2021-11-01

**Correctness:** 4
**Technical Novelty And Significance:** 3
**Empirical Novelty And Significance:** 2
**Recommendation:** 6
**Confidence:** 3

**Main Review:**

Strengths:
The method implemented in the paper is innovative and exploits a nice and not-so-popular property of the Gaussian distributions. Connecting real and discrete outputs through the expectation of the indicator function is a promising idea and may have a good impact on various discrete optimization applications. The paper has an extensive set of experiments.

weaknesses:
I have two main concerns about this work. One is the computational cost of the proposed method. Probabilistic approaches to discrete optimization often suffer from high computational complexity, e.g. the likelihood is intractable. The proposed procedure is based on a different idea but it looks like the resulting optimization requires Monte Carlo sampling or ELBO approximations. The second concern is about a possible comparison with other methods for discrete optimization, e.g. STE or more recent versions of it. If the authors have some available results, these could have been put in more evidence in the experiments section.

questions:
- Would it be possible to use the method for learning a binary NN where only the weights and not the layer representations are discrete?
- Is this the first time Gaussian stability is used for learning approximations of discrete-valued functions?
- Could be the proposed method compared with other approaches for discrete optimization, e.g. STE, on both the quality of the output and the computational cost?
- How many artificial Gaussian variables are needed for a d-dimensional optimization? Is the procedure expected to be scalable for networks with millions of parameters?

**Summary Of The Paper:**

The paper describes a computational approach to discrete optimisation, based on a probabilistic regularization procedure that enforces the output of a continuous function to be quasi-discrete. The idea is to use a property of Gaussian noise described by Borell's Isoperimetric Theorem.


**Summary Of The Review:**

A very nice idea, probably a bit expansive on the computational side.

---

> ### Author Response · Authors · 2021-11-17
> **Response**
>
> We thank the reviewer for the review and try to address the concerns below.
>
> Computational complexity/Intractable Likelihood: It is true that for the probabilistic experiments the exact likelihood is intractable and the ELBO lowerbound is approximated by MC sampling. However, this is a property of the variational approximation framework used by VAE models and is not specific to our method. We have also used the method for discrete autoencoders (non-variational, section 5.6 in the update paper) and for conditional computing (section 5.5) where the setting is not probabilistic.
>
> With regard to computational complexity, we only use one extra gaussian sample per stability layer. For the categorical VAE experiment in section 5.2 we do not use a separate stability layer, so the cost is exactly the same as the baselines. In most other experiments there is only one small stability layer (usually at most a couple of neural layers) and the overall cost is very small. This is the case in the discrete autoencoder experiment in section 5.6 where the stability layer has two convolutional layers, so that the additional cost is one extra evaluation of two conv layer per data input.
> The only experiment where we use a larger number of stability layers is the conditional computing experiment (section 5.5) where we have one stability layer for each ResNet block to be switched on or off. In that experiment the cost is higher but is nevertheless tractable for the experiment.
>
> Comparisons: In response to the comment, we have added a new comparison experiment in the updated paper in section 5.2, where we compare against STE and Gumbel-Softmax for various temperatures. The main results are that using STE with stability improves the result by about ~5 nats. Using Gumbel-Softmax with stability gives a large improvement with stability when the temperature is higher (about 10 nats improvement at tau=2, about 6 nats improvement at tau=1.5). In these experiments the stability layer is just the identity function and we directly regularize the logits so the cost is the same for all methods including Gumbel-Softmax and STE.
>
> >> Would it be possible to use the method for learning a binary NN where only the weights and not the layer representations are discrete?
>
> We consider this out of the scope for this work and we haven’t done the experiments, as learning binary NNs is a very different tasks with its own challenges. That said, it is certainly an interesting direction, and it would be interesting to explore hidden gaussian weights  regularized so that they become become binary.
>
> >> Is this the first time Gaussian stability is used for learning approximations of discrete-valued functions?
>
> We are not aware of any prior work that uses stability for learning discrete representations.
>
> >> Could be the proposed method compared with other approaches for discrete optimization, e.g. STE, on both the quality of the output and the computational cost?
>
> Yes, we have added a comparison in section 5.2, see also the response above.
>
> >> How many artificial Gaussian variables are needed for a d-dimensional optimization? Is the procedure expected to be scalable for networks with millions of parameters?
>
> Assuming d is the latent space dimension and f is the stability layer, we use 2 samples and 2 evaluations of f to compute the stability objective. Since f is usually a small network (a few layers at most), the procedure is indeed scalable to large networks. In section 5.5 we experiment with a ResNet 50 for conditional computing where we have 16 dimensional Gaussians and we use 2 samples per input.

---

### Official Review · Reviewer_btCU · 2021-11-01

**Correctness:** 3
**Technical Novelty And Significance:** 3
**Empirical Novelty And Significance:** Not applicable
**Recommendation:** 5
**Confidence:** 4

**Main Review:**

This work introduces stability regularization for training neural network with discrete stochastic variables. Unlike most of existing methods for handling discrete stochastic variables, the proposed method promotes the output of continuous functions of Gaussian random variables to discrete, which leverages the nice property of the Gaussian noise stability. The authors have conducted various experiments on different benchmarks to showcase its superiority over existing methods.

It has been known that halfspaces maximize noise stability subject to a measure constraint since Borell's work in 1985. But the theorem only states about the relationship between noise stability and partitions with two parts. This can be directly related to the binary variables, but it is not obvious how this could be extended to the categorical variables (that is, partitions with more than two parts). The way that the authors handle this is to sum across dimension, which is more like treating each dimension as independent binary variables rather than a single categorical variable. This does not prevent the output from generating all zeros or multiple ones, which is no longer a categorical variable. It has been shown that the standard simplex partition fails to maximize multi-part noise stability unless all of the parts have equal measure [1, 2]. Therefore, the analogue of Borell's theorem in the multi-part partitions case seems still an open problem. The authors might want to discuss more regarding the way that they extend the two-parts result to the categorical case and how the proposed regularization (4) promotes a single one for the categorical variable.

The authors present two versions of stability regularization: one with mean centering and the other without centering. It seems natural to compare these two versions quantitatively and give recommendations for situations where one is better than the other. However, in all 6 experiments, the authors only choose to provide results for either one of the two versions without any direct comparisons between them and the choice of which one to apply in each experiment appears a bit arbitrary. On the other hand, there are a few hyper-parameters introduced for the regularization: the correlation variable, the maximum stability, and the regularization strength. The authors claim that these either do not impact the performance much or are easier to tune without any quantitative results. The most ideal case is that one set of hyperparameters can be applied to multiple problems and gives reasonably good performance for all of them. But this does not seem to be the case here. The maximum stabilities are different across applications, and the regularization strength is only mentioned for the neural relational inference experiments. Since the "hassle-free" is one of the most important features of the proposed method, it is critical to provide sensitivity analysis of the introduced parameters and demonstrate that competitive performance can be obtained without careful tuning.

The authors also discuss how the proposed regularization can be combined with the Gumbel-Softmax method, where the output is from the Gumbel-Softmax computation but the stability is computed with the vanilla softmax without adding the Gumbel noise. This creates a mismatch here: the regularization is not applied on the "real" output. The authors might want to provide the rationale behind this and why the Gumbel noise is necessary to produce the output as input for the downstream network. One benefit of combining the stability regularization and the continuous relaxations, as the authors argue, is to provide "a form of implicit temperature control" and mitigate the need to manually tune the temperature. There are quantitative results supporting the combination is better than using Gumbel-Softmax alone. But it would be more interesting to see either the less sensitive to the temperature with the combination, or the universal improvement of the combination over the Gumbel-Softmax alone for different temperatures.

It is really nice that the authors conduct various experiments across different problem domains, but they are mostly in the same flavor without providing complementary support. I would suggest move some of the experiments to the supplementary material. The saved space can be used for the sensitivity analysis mentioned above, and the detailed comparisons with existing methods handling stochastic neural networks similar to the experiments done in [3] (the same toy experiment) and the Section B.2 in the supplementary material. After all, this is a work of training stochastic neural network. It would be nicer to give a thorough comparison with methods discussed in the Related Work.

Some typos:
- In Algorithm 2, $v_1$ and $v_2$ are used in Line 2, but $y_1, y_2$ are used in Line 3.
- The error of the mean-centered stability approximation in Proposition 1 is $O(\rho_2(\rho_2 - \rho_1))$, but $O(\rho_1 (\rho_1 - \rho_2))$ in Section A in the supplementary material.

[1] S. Heilman, E. Mossel, and J. Neeman. Standard simplices and pluralities are not the most noise stable. ITCS 2015.

[2] A. De, E. Mossel, and J. Neeman. Noise stability is computatble and approximately low-dimensional. Theory of Computing, 2019.

[3] A. Pervez, T. Cohen, and E. Gavves. Low bias low variance gradient estimates for boolean stochastic networks. ICML 2020.

**Summary Of The Paper:**

The paper proposes a method for regularizing neural networks with boolean and categorical stochastic variables. The proposed stability regularization is based on the Gaussian noise stability, which is maximized by the indicator functions of half spaces. Experiments with various benchmarks show promising results comparing to existing methods.

**Summary Of The Review:**

The proposed stability regularization has nice theoretical foundation and shows good performance across different problems involving discrete stochastic variables. But I am lean to reject mainly due to the following reasons: 1) the lack of theoretical foundation for the extension to the categorical variable, 2) the insufficient quantitative results supporting the "hassle-free" characteristic of the proposed method.

---

> ### Author Response · Authors · 2021-11-17
> **Response**
>
> We thank the reviewer for the thorough review. We have updated the paper with a new experiment on categorical VAE including comparison and the effect of various parameter settings in section 5.2.
> We try to address some of the concerns in the sequel.
>
> >> It has been known that halfspaces maximize noise stability subject to a measure constraint ...
>
> We think that there is a misunderstanding here. We do not propose to learn the simplex structure by regularization in the categorical case. Instead, the simplex structure is already imposed by using the softmax function. Therefore, there is absolutely no question of getting multiple 1’s or all 0’s.
>
> Rather, what we do is to compute the stability for each of the outputs of the softmax. Indeed, the stability for each is computed separately, but it does not matter because they are coupled with softmax. By doing so, we make it ‘easier’ for softmax to output near-discrete values (with a single 1), as the stabilized outputs of the softmax are now close to the vertices of the simplex.
>
> >> The authors present two versions of stability regularization: one with mean centering and the other without centering ...
>
> We thank the reviewer for the suggestions and opportunity to sharpen our empirical insights.
>
> In the revised section 5.2, Tables 2, we include detailed ablation studies on the impact of regularization strength hyperparameter. Specifically, we train a categorical VAE with Gumbel-Softmax at fixed temperature (tau=2) and a range of regularization strength: [80,100,120,150] (Table 2). No matter the regularization strength, we observe little fluctuations, with the performance being between about 9 and 11 nats over Gumbel softmax without regularization. This leads us to conclude that the improvement remains valid across a range of  regularization coefficients.
>
> In the appendix in Section B.3, we include an ablation study for the noise hyperparameter rho. Using the same setup as the categorical VAE experiment in Section 5.2 and using Gumbel-Softmax with stability regularizaton with temperature tau=1 and rho values in {0.5,0.6,0.8,0.9} we show validation ELBO after 300 epochs. We see that all validation ELBO are within 0.1 difference, leading us to conclude that the noise parameter in this range does not influence performance.
>
> We clarify that the stability constraint hyperparameter only depends on the number of categories and does not change per experiment. For categorical experiments, it is always chosen to be 1/k for k categories, whereas for binary latent layers we always choose it to be 0.6.
>
> As for mean centering, we note that stability regularization induces a separation in the input to the stability layer by making it noise stable. Sometimes the inputs can become too far separated, which causes a slowdown in training. This can happen in the presence of flexible stability layers or flexible decoders, or with strong regularization. When the problem appears (as in our spanning tree NRI experiments), mean centering can be used to fix it. That is, mean centering is a fix for a specific problem we observed in practice, not a new method. In general, with regular VAEs mean centering is not needed, as the VAE already has a prior that exerts a balancing force to pull the representations closer. We have made a clarification in section 3.2.
>
> We add the new results and the extra clarifications in Sections 3.2, 5.2 and B.3 in the revised version of the paper.
>
>
> >> The authors also discuss how the proposed regularization can be combined with the Gumbel-Softmax method...
>
> Stability regularization is a method that produces more accurate discrete outputs, however, this is not always the only requirement in an experimental setting. For instance, for discrete variational autoencoders we are also required to compute the sampling distribution (so that we can compute KL divergences). For such experiments, we use stability regularization combined with Gumbel-Softmax or Straight-Through. By contrast, for the clustering experiment (where we have a single categorical variable), we compute individual probabilities empirically and use those for the KL divergence computation.
>
> Regarding the effect of regularization on Gumbel-Softmax for various temperatures, we provided in the question above a detailed ablation study. We also include in section 5.3 of the updated paper a discussion of the effect of regularization on Gumbel-Softmax for various temperatures. We show that stability regularization improves Gumbel-Softmax in the high temperature regime significantly (~10 nats at tau=2), which supports our assertion that stability regularization performs implicit temperature control.
>
> We thank the reviewer again for the review and for the suggestions for experiments.

---

> > ### Comment · Reviewer_btCU · 2021-11-18
> > **Further questions**
> >
> > Thanks for the detailed response and new sensitivity analysis results . But I still have some further questions.
> >
> > First, I still have difficulty understanding how the extension of Theorem 1 to the categorical case works. As the authors acknowledged, one the contributions is the "theoretical novelty", but Theorem 1 only supports the binary cases not the categorical ones. The proposed stability regularization takes care of each element of the categorical variable independently (despite the underlying coupling of the softmax function), therefore would not prevent from generating all zeros or multiple ones. The authors argued that "there is absolutely no question of getting multiple 1’s or all 0’s". It would be much better to provide the theoretical analysis which supports this claim even just for the softmax function that the maximization of the stability regularization gives the indicator function of some simplex structure. The binary case is nicely backed with the theory, but I do not think the extension to the categorical case is straightforward.
> >
> > Second, regarding the comparison between the two versions of the stability regularizations. Thanks the authors for the clarification that mean-centering is a fix of a specific problem. But unfortunately, still no quantitative comparisons are provided in the latest version. When there is no such specific problem, would the mean-centering version perform much worse? Otherwise, why not just using the mean-centering version for all applications?
> >
> > Last, the combination with the Gumbel-Softmax method. Thanks for the new quantitative results showing the "implicit temperature control" of the proposed stability regularization. But the authors did not address the mismatch that the regularization is not applied on the "real" output and the issue behind applying it on the Gumbel-Softmax activations.

---

> > > ### Author Response · Authors · 2021-11-19
> > > **Response**
> > >
> > > Thank you for the response. We address the remaining question below.
> > > >> First, I still have difficulty understanding how the extension of Theorem 1 to the categorical case...
> > >
> > > In the categorical case, the output of a stability layer is the output of a softmax so it is not possible to get multiple 1’s or all 0’s.
> > > Assuming f(x) is some network layer producing d logits, the output of the categorical stability layer is computed as
> > > y_i(x) = softmax(f(x))_i, and sum_i y_i(x) = 1.
> > >
> > > y(x) cannot have multiple 1’s or all 0’s since it is the output of a softmax.
> > > The stability is computed and summed as S = sum_i E_{x,x’}[ y_i(x)y_i(x’)], where x,x’ are correlated Gaussian pairs, and we optimize for loss(y)+S.
> > > That is, the categorical outputs y  always sum to 1, and over training the stability term S guides the model so that they are closer to ‘discrete’.
> > > By the end, the model learns to return softmax outputs y_i, such that one of them is very close to 1, and all others close to 0.
> > >
> > > We would like to clarify that we do not claim any theoretical novelty in the application to categorical variables and, in particular, do not propose a new extension of Theorem 1. We treat the problem as one of simultaneously optimizing the stability for each output of a function, with a d-dimensional output, which have shared parameters (the network parameters) and noise variables. The final activation in the case of categorical variables is a d-way softmax.
> > >
> > > This is somewhat similar to the case of a neural network layer with d sigmoid outputs, where we also simultaneously optimize stability for each of d outputs that have shared parameters. The difference in the case of categorical variables is that the outputs are normalized to sum to 1.
> > >
> > > We add here that as a restricted case this is sound for the 2-way softmax. Since here y_2(x) = 1-y_1(x) and if we maximize y_1’s stability so that it approaches a halfspace indicator then y_2 also is a halfspace indicator of the complementary space. y_2 then would also maximize stability since it is a halfspace indicator.
> > >
> > > >>Second, regarding the comparison between the two versions of the stability regularizations…
> > >
> > > Mean-centering is appropriate in specific cases where the input to the stability network can saturate due to over-regularization, usually with a flexible stability layer and/or decoder. In general, when with small stability layers, mean-centering has worse performance than standard stability regularization and also involves more evaluations of the stability network.
> > > We experiment with the same categorical VAE setup as in section 5.2. We train the model with stability regularization (both with and without mean-centering) with Straight-Through and Gumbel-softmax (at tau=2). For Straight-Through without stability regularization we obtain a validation ELBO of 116.7; with stability regularization (no centering) 111.9, and with mean-centering 112.1 which is on-par with the performance without centering. For Gumbel-softmax (at tau=2) without regularization we obtain a validation ELBO of 132.4; without centering we get 122.7 and with centering we get 130.8 which is only marginally better than the baseline. As expected, with VAE mean centering is not needed, but it does not harm over the baseline either.
> > >
> > > >> Last, the combination with the Gumbel-Softmax method. Thanks for the new quantitative results showing the "implicit temperature control" of the proposed stability regularization...
> > >
> > > In our experience adding Gumbel noise in the stability computation did not lead to a noticeable difference.
> > > To verify this we performed another experiment with categorical VAE using stability with Gumbel-Softmax and stability computation with Gumbel noise. The results are given in table 8 in the appendix in the updated paper in Section B.3.2. The conclusion is that Gumbel noise can help improve performance at higher temperatures (tau=1.5, 2) by about 2 nats. For lower temperatures (tau=0.1, 0.5, 1), stability with Gumbel noise is on-par or marginally worse (about half a nat) than stability without Gumbel noise. Overall we conclude that the performance difference is marginal.

---

### Official Review · Reviewer_k6BN · 2021-11-03

**Correctness:** 3
**Technical Novelty And Significance:** 2
**Empirical Novelty And Significance:** 2
**Recommendation:** 5
**Confidence:** 4

**Main Review:**

Overall, I think the proposed new method is interesting and worth further study, but I also feel that the contribution of the submission is somewhat limited.

In particular, I have several concerns about the submission.

1. I think the study is not thorough enough. Though we see some performance improvement, can we compare the proposed method against Gumbel-softmax at a more detailed level. For example, Gumbel-softmax uses temperature to control the extremeness of function values. I think the corresponding hyper-parameter in the proposed method is the strength of regularization. Can we do a side-by-side comparison between the two hyper-parameters. For example, checking the performance vs temperature, and the performance vs regularization strength.

My hypothesis is that the temperature value is hard to tune while the regularization strength is easier to tune. The performance difference might be from this reason.

In general, I think the optimization involving discrete variables is hard, and the backpropagation through these variables does not behave. I don't feel the proposed method is changing the property.

2. The performance improvement does not seem to be consistent. Sometimes we see performance drops. Is the experiment in 5.3 a good comparison? The proposed method uses 20 clusters while the first two baselines use 30 clusters. More clusters should leads to higher performance, right? The authors may want to increase the number of clusters and check the performance. How do you get the performance of SB-VAE? I don't find it in the original paper.

3. Which experiments use mean-centering? Do you see a performance improvement by using mean-centering? I don't see a reason for mean centering in experiments. I feel that the model will learn the function and decide an underlying mean even without mean-centering.






**Summary Of The Paper:**

The paper proposes an interesting regularization method for encouraging function values to be discrete. The main idea is very interesting: with two correlated random vectors, the function is encouraged to maximize the correlation of outputs from the two vectors. The function that maximizes the correlation is the indicator function of a half-space. Combining this objective and original training objectives, a learning model can output random values that are near 0 or 1. The model plays the same role as the Gumbel-softmax trick.

The submission has done extensive experiments comparing the proposed technique against Gumbel-softmax. The results indicate that the new technique improves the performance in several learning tasks, though the improvement is not consistent.

**Summary Of The Review:**

Overall I think the technique is interesting, but I also feel the overall contribution of the submission is somewhat limited.

---

> ### Author Response · Authors · 2021-11-17
> **Response**
>
> We thank the reviewer for the kind words, the detailed review, and the opportunity to address any concerns.
>
> >> 1. I think the study is not thorough enough. Though we see some performance improvement, can we compare the proposed method against Gumbel-softmax at a more detailed level...
>
> Thank you for the suggestion. We have added a new comparison experiment to the paper on VAE with categorical variables, and compare methods with and without stability regularization for a range of hyperparameters. The experiment is in section 5.2 in the updated paper (changes are colored in red).
>
> We summarize the experiment here: We compare Gumbel-Softmax with and without stability regularization for a range of temperatures in [0.1,0.5,1,1.5,2]. The experiment (Table 1) shows that Gumbel-Softmax is much more sensitive to increasing temperatures without stability regularization, where the gap widens the higher the temperature is set. When setting tau=2, stability regularization brings an improvement of ~10 nats, for instance.  This provides credence to our hypothesis that stability regularization has the effect of temperature control.
>
> Next, to test the hypothesis that the regularization strength acts like a temperature we train the model with Gumbel softmax and stability regularization at fixed temperature (tau=2) and a range of regularization strength: [80,100,120,150] (Table 2). No matter the regularization strength, we observe little fluctuations, with the performance being between about 9 and 11 nats over Gumbel softmax without regularization. This leads us to conclude that the regularization coefficient serves a different function than the temperature. In our view the regularization coefficient is like any other regularization term parameter that balances between the main objective and the regularization objective.
>
>
> >> In general, I think the optimization involving discrete variables is hard, and the backpropagation through these variables does not behave. I don't feel the proposed method is changing the property.
>
> Part of the reason the optimization misbehaves is that especially with continuous relaxation the distribution parameters (say logits) tend to cluster together. This is usually handled with a manual temperature scale, where a small temperature in the softmax induces a larger separation. Unfortunately, as motivated in the paper and in the experiment above, methods like Gumbel-Softmax are quite sensitive to the choice of temperature. Our method handles this by exploiting the fact that in Gaussian spaces, it is guaranteed that increasing the stability of Gaussian functions yields half spaces, and we can use these half spaces to improve the separation. This separation improves the backprop through discrete variables.
>
> >> 2. The performance improvement does not seem to be consistent. Sometimes we see performance drops. Is the experiment in 5.3 a good comparison? The proposed method uses 20 clusters while the first two baselines use 30 clusters.
>
> It is not necessarily true that increasing the number of clusters increases performance. Our method involves backprop through a discrete categorical variable which becomes harder as the number of categories increases. The baselines we compare against do not use backprop through discrete stochastic variables and the GMVAE method [2], in fact, marginalises over the categorical variable. The aim of the comparison is to show that similar performance can be achieved by using a very simple two layer model and backprop through categorical variables instead of marginalisation.
>
> The SB-VAE result is given in the SB-VAE paper [1] in Figure 4 (a). We use the best result reported there.  We further note that in the SB-VAE paper clustering performance does not increase with increasing the number of clusters (error of 8.65 for 5 and 8.90 for 10 clusters).
>
> Mean Centering: Stability regularization induces a separation in the input to the stability layer by making it noise stable. Sometimes the inputs can become too far separated which causes a slowdown in training. This can happen in the presence of flexible stability layers or flexible decoders or with strong regularization. When the problem appears (as in our spanning tree NRI experiments), mean centering can be used to fix it.
>
> That is, mean centering is a fix for a specific problem we observed in practice, not a new method. In general, with regular VAEs mean centering is not needed, as the VAE already has a prior that exerts a balancing force to pull the representations closer. We have made a clarification in section 3.2.
>
>
> [1] Nalisnick and Smyth, Stick-Breaking Variational Autoencoders
>
> [2] Dilokthanakul et al. Deep Unsupervised Clustering with Gaussian Mixture Variational Autoencoders

---

### Author Response · Authors · 2021-11-17
**Acknowledgement**

We thank all reviewers for recognizing the theoretical novelty of our work in modelling discrete random variables via the lens of stability in Gaussian functions, as well as appreciating the extensive experimental verification. We understand that reviewers would appreciate more detailed ablation studies to prove the robustness of our model, which we happily provide with expected positive results.

We hope that our responses will alleviate any remaining concerns and convince you to recommend acceptance. We welcome any further opportunities for clarification.

---

### Decision · Program_Chairs · 2022-01-20

**Decision:**

Accept (Poster)

**Comment:**

The paper introduces a method to train neural networks based on so-called stability regularisation. The method encourages the outputs of functions of Gaussian random variables to be close to discrete and does not require temperature annealing like the Gumbel Softmax. All reviewers agreed that the proposed method was novel and of interest. The authors conducted extensive experiments. They also adequately addressed the concerns raised by the reviewers (e.g., theoretical foundation and computational cost).